# Immunogenicity and Protective Efficacy of Dose-Sparing Epigraph Vaccine against H3 Swine Influenza A Virus

**DOI:** 10.3390/vaccines12080943

**Published:** 2024-08-22

**Authors:** Erika Petro-Turnquist, Adthakorn Madapong, David Steffen, Eric A. Weaver

**Affiliations:** 1Nebraska Center for Virology, School of Biological Sciences, University of Nebraska-Lincoln, Lincoln, NE 68583, USA; epetro-turnquist2@huskers.unl.edu (E.P.-T.); amadapong2@unl.edu (A.M.); 2Nebraska Veterinary Diagnostics Center, Lincoln, NE 68583, USA; dsteffen1@unl.edu

**Keywords:** Swine influenza A virus, Epigraph, dose-sparing, Cluster IV(A), Cluster 1, Cluster 2010.1 “human-like”

## Abstract

Swine influenza A virus (IAV-S) is a highly prevalent and transmissible pathogen infecting worldwide swine populations. Our previous work has shown that the computationally derived vaccine platform, Epigraph, can induce broadly cross-reactive and durable immunity against H3 IAV-S in mice and swine. Therefore, in this study, we assess the immunogenicity and protective efficacy of the Epigraph vaccine at increasingly lower doses to determine the minimum dose required to maintain protective immunity against three genetically divergent H3 IAV-S. We assessed both antibody and T cell responses and then challenged with three H3N2 IAV-S derived from either Cluster IV(A), Cluster I, or the 2010.1 “human-like” cluster and assessed protection through reduced pathology, reduced viral load in the lungs, and reduced viral shedding from nasal swabs. Overall, we observed a dose-dependent effect where the highest dose of Epigraph protected against all three challenges, the middle dose of Epigraph protected against more genetically similar IAV-S, and the lowest dose of Epigraph only protected against genetically similar IAV-S. The results of these studies can be used to continue developing a broadly protective and low-dose vaccine against H3 IAV-S.

## 1. Introduction

Influenza A virus (IAV) is an acute respiratory pathogen within the family *Orthomyxoviridae.* IAV genomes are composed of eight segments encoding at least 10 proteins (PB1, PB2, PA, HA, NP, NA, M1, M2, NS1, and NS2/NEP) [1]. The error-prone RNA polymerase leads to rapid mutation of the viral proteins (antigenic drift), and the segmentation of the viral genome can result in reassortment between multiple strains during co-infection (antigenic shift). Swine are susceptible to avian, human, and other swine IAV due to the expression of both α-2,3 and α-2,6-linked sialic acid receptors in the respiratory tract [2,3]. IAV in swine (IAV-S) is often characterized by a rapid onset of high fever, reproductive failure, inappetence, respiratory distress, and coughing. IAV-S infects swine at all stages of pork production and consequently imposes a significant economic burden on the pork industry. These economic losses can be as high as USD 26.10/pig [4] and are due to reduced weight gain [5] and decreased piglet births per sow in breeding herds endemically infected with IAV-S [6]. Further, swine are considered a significant intermediate reservoir for zoonotic, potentially pandemic IAV and pose a substantial threat to public health [7]. Consequently, increased efforts have focused on appropriate surveillance and control of IAV-S in swine.

Currently, three subtypes of IAV-S circulate in worldwide swine populations: H1N1, H1N2, and H3N2 [8,9]. The worldwide seroprevalence of IAV-S in swine ranges from 49.9% to 72.8%, depending on the region and vaccination status of the herd [10]. Influenza viruses of the H1 subtype exclusively circulated in swine populations until 1998, when a triple reassortant H3N2 was detected in North America [11]. This novel IAV-S contained genes from human (HA, NA, and PB1), swine (NS, NP, and M), and avian (PB2 and PA) lineages and was designated as the Cluster I H3 (or 1990.1) IAV-S. This clade subsequently evolved into Cluster IV (1990.4), which later diverged into the subclades A-F (1990.1a-f) [8]. Additionally, the repeated reverse zoonotic transmission of human-to-swine H3N2 has led to the development of novel endemic clades of “human-like” H3 IAV-S. In 2012, the 2010.1 clade was established [12] and currently makes up ~42.9% of reported IAV-S infections in U.S. swine populations [13]. In 2016, another reverse zoonotic transmission from humans to swine established the 2010.2 clade [14], though this clade is less prevalent than the 2010.1 clade [13]. The expansive genetic diversity and distinct antigenicity of the H3 IAV-S subtype represent a significant challenge in developing effective methods of IAV-S control in swine.

The most common method of controlling IAV-S is through strict biosecurity measures [15] and vaccination [16]. Current commercial vaccines include whole-inactivated virus (WIV) vaccines, replicon particle (RP) vaccines, and previously live attenuated virus (LAIV) vaccines. While WIV vaccines are the most common platform used against IAV-S, WIV vaccines are typically hallmarked by low cross-protection against antigenically distinct IAV-S and antibody-restricted immune induction. RP vaccines can induce heterologous cross-reactive immunity against IAV-S but can be expensive to produce. Both WIV and RP vaccines can be made autogenously; however, this method often requires labor-intensive laboratory techniques for diagnostics, virus isolation, and vaccine production that can be time- and cost-prohibitive. Further, autogenous vaccines are not required to be tested for efficacy prior to use. In 2017, a LAIV vaccine containing two subtypes of IAV-S (H1N1 strain A/swine/Minnesota/37866/1999 and H3N2 strain A/swine/Texas/4199-2/1998) was licensed for use in U.S. swine populations. This LAIV was shown to induce improved heterologous protection compared to WIV vaccines. However, subsequent surveillance revealed reassortment between the LAIV vaccine and endemic field strains in U.S. swine populations [17]. Consequently, the LAIV was removed from commercial use in 2020. Given the drawbacks of current vaccines against IAV-S, there is a need for improved vaccination strategies against IAV-S in swine. An ideal vaccine against IAV-S should be easy and inexpensive to mass-produce, induce broadly cross-reactive immunity, and protect against a wide variety of genetically relevant IAV-S.

Our previous research has evaluated the immunogenicity and protective efficacy of the universal vaccine immunogens against IAV [18,19,20,21,22,23]. The Epigraph is a computational platform that maximizes potential T and B cell epitopes incorporated in a vaccine immunogen design to induce broadly cross-reactive immunity against pathogens with high genetic diversity [24]. We have recently demonstrated that Epigraph immunogens can induce broadly cross-protective antibodies against both human and swine H3 IAV in swine and that the Epigraph immunogens significantly outperform a wild-type immunogen and a commercial comparator WIV vaccine [18]. We have also recently established that these responses are maintained for 6 months after the initial vaccination and can protect against heterologous challenges in swine [20]. However, given that there are several endemic H3 clades circulating in swine populations with variable antigenic diversity, the objective of this study was to define the minimum dose required to maintain protective immunity against a broad range of H3 IAV-S. In this study, we assess the immunogenicity and protective efficacy of the Epigraph vaccine at increasingly lower doses and define the minimum dose required to protect swine against three H3 IAV-S challenges derived from Cluster IV(A), Cluster I, and the 2010.1 Cluster.

## 2. Materials and Methods

### 2.1. Ethics Statement

All experiments in swine were performed in accordance with the animal welfare requirements of the University of Nebraska–Lincoln (UNL). All experimental protocols were approved by the UNL Institutional Animal Care and Use Committee (IACUC; Protocol number 2167), and swine were housed in the Life Sciences Annex building on the UNL campus under the Association for Assessment and Accreditation of Laboratory Animal Care International guidelines. The health of all swine was consistently monitored by an accredited veterinarian, and samples from swine that demonstrated instances of clinical disease were collected and sent to the Nebraska Veterinary Diagnostic Center.

### 2.2. Cells and Viruses

Madin Darby canine kidney cells–London line (MDCK-Ln) cells were cultured in complete Dulbecco’s Modified Eagle Medium (DMEM; Cytiva; Marlborough, MA, USA) supplemented with 5% fetal bovine serum and 1% penicillin–streptomycin (P/S; Cytiva; Marlborough, MA, USA). Cells were maintained at 37 °C and 5% CO_2_. The following swine H3N2 influenza viruses were obtained from the Biodefense and Emerging Infectious Diseases Repository: [Abbreviation: GISAID Isolate ID; Cluster Designation] A/swine/Ohio/09SW73E/2009 (OH/09; EPI_ISL_133799; Cluster-IV) and A/swine/Ohio/11SW87/2011 (OH/11; EPI_ISL_133873; Cluster-IV(A)). The following viruses were obtained from the U.S. Department of Agriculture Swine Surveillance Influenza A virus isolate repository: A/swine/Colorado/23619/1999 (CO/99; EPI_ISL_1751; Cluster-II), A/swine/Wyoming/A01444562/2013 (WY/13; EPI_ISL_136647; Cluster-IV(A)), A/swine/Minnesota/A01270872/2012 (MN/12; EPI_ISL_132375; Cluster-IV(B), A/swine/Indiana/A01202866/2011 (IN/11; EPI_ISL_127867; Cluster-IV(C)), A/swine/Texas/A01785781/2018 (TX/18; EPI_ISL_366850; Cluster-2010.1). The following viruses were generous gifts from our collaborators: A/swine/Texas/4199-2/1998 (TX/98; EPI_ISL_94059; Cluster-I) and A/swine/Kansas/11-110529/2011 (KS/11; EPI_ISL_100252; Cluster-IV(F)) from Dr. Hiep Vu and Dr. Wenjun Ma, respectively. All viruses were grown in specific pathogen-free (SPF) embryonated eggs for 72 h, and the allantoic fluid was collected and stored at −80 °C. Virus stocks were quantified by the hemagglutination assay and TCID_50_ for use in downstream assays.

### 2.3. Animal Immunization, Challenge, and Sampling

We performed three experiments using a total of 75 outbred, aged-matched male and female Yorkshire pigs purchased from Midwest Research Swine. All pigs were screened for prior swine influenza A virus exposure and confirmed seronegative for influenza antibodies before immunization. Pigs were randomly allocated into vaccine groups and allowed to acclimate for 5–7 days prior to vaccination. Each vaccination group was housed in a separate BSL2+ isolation room for the full duration of the experiments. In experiment 1, pigs (*n* = 5/vaccine group) were intramuscularly vaccinated with 10^11^vp, 10^10^vp, or 10^9^vp of Epigraph vaccine, FluSure XP, or DPBS as a sham vaccination. Three weeks later, blood was collected from all pigs through the anterior jugular puncture, and all pigs were boosted with the respective vaccine. Two weeks after the boost immunization, pigs were bled and then challenged with a split intranasal (2.0 mL) and intratracheal inoculation (2.0 mL) of 10^6^ TCID_50_/mL of A/swine/Ohio/11SW87/2011 (H3N2; Cluster—IV(A)) [25]. The inoculum was administered with 2 mL of 2.5 × 10^5^ TCID_50_/mL of virus instilled in the trachea and 1 mL of 2.5 × 10^5^ TCID_50_/mL instilled in each nostril for a total dose of 1 × 10^6^ TCID_50_ [25]. Nasal swabs were collected daily after the challenge, and 5 days post-challenge, all pigs were euthanized to evaluate viral titers in the bronchioalveolar lavage (BAL), viral antigen, and histopathological lesions in the trachea and lungs. In experiment 2, all pigs were vaccinated, and blood samples were collected as described, and the pigs were challenged with 10^6^ TCID_50_ A/swine/Texas/4199-2/1998 (H3N2; Cluster—I). In experiment 3, pigs were vaccinated, blood samples were collected as described, and the pigs were challenged with 10^4.5^ TCID_50_ A/swine/Texas/A01785781/2018 (H3N2; Cluster—2010.1). Differences in the challenge doses were due to variations in pathologies induced by the various strains. In experiment 3, *Haemophilus parasuis* were detected in 9 out of 25 pigs at necropsy. Nine pigs had chronic fibrous pleural or, more often, pericardial adhesion. Pig 75 had active inflammation at autopsy, and H. parasuis was cultured only from that pig. The adhesive lesions are typical of those described for a resolved H. parasuis infection. Clinically apparent pericarditis or pleuritis symptoms were not observed during the study in the 8 pigs with inactive lesions. An unpaired parametric *t*-test was performed on day 35 (after a boost immunization prior to the challenge) to compare the T cell responses of pigs positive for H. parasuis to pigs negative for H. parasuis and determined that there was no statistically significant effect of the H. parasuis infection on the T cell responses to the vaccines. It is suspected that the infections had resolved prior to study initiation. No substantial clinical disease associated with pathogenic H. parasuis infection was noted during experiment 3. No substantial clinical disease associated with pathogenic *H. parasuis* infection was noted during experiment 3. All challenges were performed under telazol (Zoetis; Parsippany, NJ, USA), zolazepam (Zoetis; Parsippany, NJ, USA), ketamine (Zoetis; Parsippany, NJ, USA), and xylazine (Vet One; Las Vegas, NV; USA)-induced anesthesia.

### 2.4. Antibody Assay

Sera samples were obtained from whole blood collected in BD Vacutainer Separator Tubes (Becton Dickinson; Franklin Lakes, NJ, USA) after the prime and boost immunizations. Serum was used for the hemagglutination inhibition (HI) assay according to previously described methods [26]. Briefly, sera were treated with receptor-destroying enzyme (RDE; Denka Seiken; Chuo-ku, Tokyo; Japan) at a 1:3 ratio (sera:RDE) at 37 °C for 18 h. Sera samples were then heat-inactivated at 56 °C for 1 h and diluted to a final 1:10 dilution in DPBS. Sera were serially diluted two-fold in a 96-well V-bottom plate, and 4 hemagglutinating units of the respective virus were added to all wells and then incubated at room temperature for 45 min. After incubation, 50 μL of 0.5% chicken (rooster) red blood cells was added to all wells, and hemagglutinating patterns were read after 30 min.

### 2.5. T Cell Analysis

Peripheral blood mononuclear cells (PBMCs) were isolated from whole blood collected in BD Vacutainer tubes (Becton Dickinson; Franklin Lakes, NJ, USA). PBMCs were isolated as described previously [20]. PBMCs were analyzed for total IFN-γ-secreting T cell responses against three human H3 influenza A viruses. Overlapping peptide arrays spanning the entire length of the HA protein of A/Uruguay/716/2007 (H3N2; Clade—3C.2) (NR-18968), A/New York/384/2005 (H3N2; Clade—3C.3a) (NR-2603), and A/Perth/16/2009 (H3N2; Clade—3C.3a) (NR-19266) were obtained from the Biodefense and Emerging Infectious Diseases Repository and used for T cell analysis. Peptide pools were developed for each HA protein for the re-stimulation of swine PBMC. Polyvinylidene 96-well difluoride-backed plates (MultiScreen-IP; Millipore; Billerica, MA, USA) were coated overnight at 4 °C with 5 µg/mL of anti-swine IFN-γ mAb (pIFNγ-I; Mabtech; Cincinnati, OH, USA). All wells were washed and blocked with RPMI containing 10% FBS and 1% P/S for 2 h before stimulating single-cell suspensions of swine PBMC with peptide pools (5 µg/mL/peptide) overnight at 37 °C and 5% CO_2_. PBMCs were also stimulated with Concavalin A (ConA; 5 µg/mL) or RPMI to serve as positive and negative controls, respectively. After overnight incubation, all wells were washed six times with DPBS containing 0.1% Tween-20 (DPBS-T), then 1 µg/mL of biotinylated anti-porcine IFN-γ mAb (P2C11; Mabtech; Cincinnati, OH, USA) was added to all wells and incubated at room temperature for 1 h. Plates were then washed six times with DPBS-T, and a 1:1000 dilution of streptavidin-alkaline phosphatase conjugate (Mabtech; Cincinnati, OH, USA) was added, incubated for 45 min, washed, and then developed with BCIP/NBT (Plus) alkaline phosphatase substrate (Thermo Fisher; Rockford, IL, USA). Plates were developed until spots formed in the positive control ConA stimulation wells, and then development was stopped by washing wells several times with ddH_2_O. Total numbers of spot-forming units (SFUs) were determined using an automated ELISpot plate reader (Cellular Technology Ltd.; Cleveland, OH, USA) and represented as the number of SFUs per 10^6^ cells.

### 2.6. Quantification of Viral Load

The viral load in daily nasal swabs collected during the challenge and BAL samples collected after the necropsy was determined by TCID_50_. Nasal swabs and BAL samples were diluted 1:10 in a sterile 96-well U-bottom tissue-culture dish and serially diluted 10-fold. MDCK-Ln cells (2 × 10^4^ cells) were added to each well, and plates were incubated overnight at 37 °C with 5% CO_2_. The next day, plates were washed twice with sterile DPBS, and DMEM with 2 µg/mL of TPCK-trypsin was added to all wells. The plates were incubated at 37 °C with 5% CO_2_ for 72 h before adding 50 µL of 0.5% chicken red blood cells and reading hemagglutination patterns after 1 h.

### 2.7. Lung Pathological Analysis

Sections of lung and trachea samples were collected 5 days after the infection. Animals were euthanized with an overdose of sodium pentobarbital Fatal Plus (Vortech; Dearborn, MI, USA), and one-centimeter-thick tissue samples were collected from the middle trachea (approximately 5–8 cm above the right tracheal bronchus), apical, middle, and caudal lung were collected and placed in 10% neutral buffered formalin. All lung samples were collected from the right lung. After formalin fixation for 72-h, lung and trachea tissue were sectioned at 4–5 µm, stained with hematoxylin and eosin (H&E) and scored by an ACVP-certified pathologist according to a previously described scoring system [27]. Tracheas were scored as 0, normal; 1, focal inflammation with the presence of cilia; 2, diffuse inflammation and multifocal cilia loss; and 3, widespread inflammation and ciliary loss. Lung sections were scored from −3 based on the severity of necrotizing bronchitis and bronchiolitis, suppurative bronchitis and bronchiolitis, peribronchiolar lymphocytic cuffing, and alveolar septal inflammation. A composite score was computed using the sum of the four individual scores, and an average group composite score was used for statistical analysis. Immunohistochemistry (IHC) was performed against a conserved IAV nucleoprotein antigen using a rabbit anti-influenza A virus NP antibody (Invitrogen; Carlsbad, California, United States). Lung and trachea sections were fixed and pretreated with Ultra Cell Conditioning Solution (ULTRA CC1) for 1 h before incubation with rabbit anti-influenza virus NP antibody for 30 min (1:1000; Invitrogen; Carlsbad, CA, USA; #PA5-32242). Viral NP antigens were visualized using the UltraView Universal Alkaline Phosphatase Red Detection Kit (Roche; Mannheim, Germany; 760-501) and counterstained with hematoxylin and bluing reagent (Roche; Mannheim, Germany).

### 2.8. Statistical Analysis

All statistical analysis was carried out using GraphPad Prism v10.2.2. Antibody titers were log_2_ transformed for statistical analysis. Data are presented as means with standard error measurement (SEM). An area under the curve (AUC) was completed for daily nasal swabs to assess statistical analysis over the course of the infection. Sex-based differences were analyzed for antibody and T cell responses by categorizing female-specific and male-specific responses per dose, per virus, and per timepoint. Female- and male-specific differences were compared using unpaired *t*-tests. Differences between vaccine groups with respect to antibody, T cell, viral load, and composite histopathological analysis were analyzed by one-way analysis of variance (ANOVA) with Tukey’s multiple comparisons follow-up test. A *p*-value < 0.05 was considered statistically significant (ns: not significant; * *p* < 0.05; ** *p* < 0.01; *** *p* < 0.001; **** *p* < 0.0001).

## 3. Results

### 3.1. Antibody Responses in Dose Sparing

Our previous research has demonstrated that the Epigraph platform can induce broadly cross-reactive antibody responses against swine and human H3 influenza A viruses [18]. We have also shown that these responses are rapid-onset and can remain durable for up to 6 months after the initial vaccination to provide protection against heterologous challenges [20]. Consequently, in this study, we wanted to assess the breadth of protection of the Epigraph vaccine in a dose-sparing study. We immunized pigs with 10^11^vp, 10^10^vp, or 10^9^vp of the Epigraph vaccine and compared these responses to a commercial vaccine, FluSure XP. FluSure XP is an adjuvanted whole inactivated virus vaccine commonly used in the United States and includes a strain from Cluster IV(A), A/swine/North Carolina/A01270394/2012, and a strain from Cluster IV(B), A/swine/Minnesota/A01270872/2012. We used an HI titer of ≥1:40 (5.32 log_2_) as a threshold of protective antibody titers because previous research has demonstrated that HI antibody responses at or above these levels correspond to a 50% reduction in the risk of influenza infection [28,29]. The assessment of antibody responses after a single immunization showed broadly cross-reactive antibody responses in a dose-dependent manner (Figure 1A). Notably, with the exception of the antibody responses against KS/11, we did not observe any statistically significant differences in antibody responses when comparing 10^11^vp and 10^10^vp, indicating that reducing the dose by 10-fold does not significantly impact immune induction. Pigs vaccinated with 10^11^vp induced significantly higher HI antibody responses against TX/98, CO/99, OH/09, WY/13, KS/11, and TX/18 than pigs immunized with 10^9^vp. In contrast, pigs vaccinated with 10^10^vp only induced significantly higher HI antibody responses than 10^9^vp against WY/13. After a single immunization, pigs vaccinated with 10^11^vp or 10^10^vp induced protective HI antibody responses (≥40; 5.32 log_2_) against 4 out of the 8 viruses tested (50% protection), while pigs vaccinated with 10^9^vp only induced protective HI antibody responses against 1 out of the 8 viruses tested (~12.5% protection). Pigs vaccinated with the commercial comparator vaccine, FluSure XP, showed protective antibody titers only against closely matched viruses, WT/13 and MN/12, after a single immunization. We observed that these responses increased after a boost immunization and again followed a pattern of significantly lower HI antibody responses in the lowest dose group, 10^9^vp (Figure 1B). In agreement with previous results, pigs immunized with FluSure XP demonstrated a significant boost after a second immunization and had significantly higher HI antibody responses against TX/98, CO/99, WY/13, MN/12, IN/11, and TX/18 compared to the different doses of the Epigraph vaccine (Figure 1B). After a second immunization, pigs vaccinated with 10^11^vp or 10^10^vp induced protective HI antibody responses against 6 out of 8 viruses tested (75% protection), while pigs vaccinated with 10^9^vp induced protective titers against 4 out of 8 viruses tested (50% protection) (Figure 1B). We further analyzed the induction of sex-specific antibody responses by comparing antibody responses between males and females in each vaccine dose. After the prime and boost vaccinations, we observed little-to-no significant differences in antibody induction between males and females against any virus at either the prime or boost immunizations (Appendix A). These similarities in responses were also observed in the FluSure XP group. This is important because an ideal vaccine against IAV-S will induce similar levels of protection in both male and female pigs and should be able to be administered to all swine populations despite differences in sex.

### 3.2. T Cell Responses in Dose Sparing

Cross-reactive T cells play an important role in clearing influenza-infected cells during an active viral infection [30,31,32,33]. Therefore, we wanted to evaluate the induction of cross-reactive T cell responses after vaccination through an interferon-γ enzyme-linked immunospot (IFN-γ ELISpot) assay. We assessed cross-reactive T cell responses against three divergent H3N2 strains using an interferon-γ enzyme-linked immunospot (IFN-γ ELISpot) assay. PBMCs were collected after a prime and boost immunization and compared among groups. After a single immunization, pigs immunized with 10^11^vp and 10^10^vp demonstrated significantly higher T cell responses against Uruguay/07 (Figure 2A), New York/05 (Figure 2B), and Perth/09 (Figure 2C) compared to FluSure XP-immunized pigs. In contrast, pigs immunized with 10^9^vp showed modest T cell induction that was not significantly higher than FluSure XP immunized pigs (Figure 2). Interestingly, while pigs immunized with 10^11^vp showed trends of higher T cell responses after a boost immunization, pigs immunized with 10^10^vp and 10^9^vp did not demonstrate boosting after the second immunization (Figure 2). Females immunized with a single dose of 10^11^vp were shown to induce significantly higher IFN-γ secreting T cell responses against Uruguay/07 compared to males, though both males and females had similar responses against all strains tested after a boost immunization (Appendix A). Similarly, female pigs immunized with a single dose of 10^10^vp showed significantly higher levels of IFN-γ-secreting T cell responses against New York/05 than male pigs, but these responses were similar after the boost immunization (Appendix A). Finally, after the second immunization, male pigs vaccinated with 10^9^vp showed significantly higher levels of IFN-γ T cell responses against Uruguay/07 and Perth/09 than female pigs. Notably, female pigs showed trends of slightly lower T cell responses, while males maintained similar levels of T cell responses after the boost immunization (Appendix A). No significant differences were observed between female or male pigs in the FluSure XP immunization group (Appendix A).

### 3.3. Protection against Cluster IV(A) Swine Influenza a Virus

Since 2009, 41.94% of all H3 IAV-S infections in U.S. swine populations have belonged to cluster IV(A) [13]. Consequently, it is imperative that a broadly protective IAV-S vaccine provide optimal protection against this cluster of IAV-S and others. To assess protection against challenge, swine were experimentally infected with 10^6^ TCID_50_ of a representative Cluster IV(A) IAV-S isolate, A/swine/Ohio/11SW87/2011 (98.2% sequence identity to the closest matched Epigraph; 99.3% sequence identity to the closest matched FluSure XP strain). Nasal swabs were collected daily, and lungs were collected after infection to assess viral load and histopathology. Pigs vaccinated with 10^11^vp, 10^10^vp, and 10^9^vp of the Epigraph vaccine and FluSure XP were completely protected from pathological damage and had no viral antigens in the lungs (Figure 3A,B). These pigs also displayed no tracheal damage or viral load in tracheal samples (Figure 3C,D). In contrast, unimmunized pigs showed severe bronchiolitis coupled with pulmonary edema and viral antigen in the lungs (Figure 3A,B), epithelial metaplasia, ciliary loss, and viral antigens in the trachea (Figure 3C,D). Composite scores of histopathology revealed that pigs immunized with any dose of Epigraph or FluSure XP had significantly reduced overall pathology compared to unimmunized pigs (Figure 3E), and no infectious virus was collected from immunized pigs (Figure 3F). Finally, all immunized pigs had significantly lower viral shedding compared to unimmunized control animals (Figure 3G and Appendix A). These data indicate that pigs vaccinated with the lowest dose (10^9^vp) of Epigraph were completely protected against a closely matched Cluster IV(A) swine influenza A virus.

### 3.4. Protection against Cluster I Swine Influenza a Virus

Cluster I H3 swine influenza A viruses were first detected in the late 1990s and have been minorly detected since the development of the Cluster IV(A-F) H3 IAV-S. However, after reassortment between a LAIV vaccine containing the A/swine/Texas/4199-2/1998 HA and field strains [17], Cluster I H3 IAV-S was steadily detected in 2018-2022 [13] and could potentially be endemic to U.S. swine populations. Consequently, we next wanted to assess protection against a heterologous Cluster I IAV-S, A/swine/Texas/4199-2/1998 (91.5% sequence identity to the closest matched Epigraph; 91.0% sequence identity to the closest matched FluSure XP strain). Pigs vaccinated with 10^11^vp and 10^10^vp were completely protected from pathological findings and viral infection in the lungs and trachea (Figure 4A–F). In contrast, pigs immunized with the low-dose Epigraph demonstrated more severe pathological damage in the lungs and trachea (Figure 4A,C) that was not statistically significantly reduced compared to unimmunized controls (Figure 4E). Pigs immunized with FluSure XP showed less severe pathological damage in the lungs and trachea compared to unimmunized control animals (Figure 4A,C,E). When a bronchoalveolar lavage was performed, immunized pigs showed slightly more variability in infectious virus titers, but all immunized pigs had significantly lower infectious viruses compared to unimmunized pigs (Figure 4F). Analysis of infectious virus collected from the nose indicated that pigs immunized with 10^11^vp and 10^10^vp of the Epigraph vaccine and FluSure XP had a shorter duration of viral shedding compared to unimmunized pigs (Figure 4G and Appendix A). However, pigs immunized with the low-dose Epigraph vaccine had more sustained viral shedding after the challenge (Figure 4G and Appendix A). These data indicate that the high-dose and medium-dose Epigraph vaccines are required to robustly protect against a highly divergent Cluster I IAV-S.

### 3.5. Protection against Cluster 2010.1 Human-like Swine Influenza a Virus

In 2012, an epidemiological analysis of U.S. swine populations detected a reverse zoonotic transmission event that led to the establishment of the 2010.1 “human-like” H3 Cluster [34]. Since this introduction, there has been a steady increase in the prevalence of IAV-S infections belonging to Cluster 2010.1 [13], highlighting the necessity to protect against this clade of IAV-S. Consequently, we assessed protection against a representative 2010.1 IAV-S isolate, A/swine/Texas/A01785781/2018 (90.8% sequence identity to the closest matched Epigraph; 87.8% sequence identity to the closest matched FluSure XP strain). Similar to the results obtained after the Cluster I challenge, pigs immunized with 10^11^vp and 10^10^vp of the Epigraph vaccine had generally reduced pathological findings in the lungs and trachea compared to unimmunized animals (Figure 5A,C). This was coupled with no viral antigen present in the lungs or trachea of pigs immunized with 10^11^vp of Epigraph (Figure 5B,C) and minimal viral antigen in the lungs of pigs immunized with 10^10^vp (Figure 5B). In contrast, pigs immunized with 10^9^vp of Epigraph demonstrated more variable pathological findings, with some evidence of mild interstitial pneumonia in 2 of the 5 pigs (Figure 5A), high viral antigen in the lungs (Figure 5B), and mild detection of viral antigen in trachea samples (Figure 5D). Pigs vaccinated with FluSure XP had some instances of bronchiolitis, though this was lower than the extensive pathological findings in the unimmunized control animals (Figure 5A). Overall, pigs vaccinated with 10^11^vp and 10^10^vp of the Epigraph vaccine or FluSure XP had trends of lower composite scores compared to unimmunized pigs, while animals in the 10^9^vp group had similar levels (Figure 5E). Notably, pigs vaccinated with 10^11^vp and 10^10^vp of Epigraph or FluSure XP also showed lower levels of infectious virus in the lungs when a bronchioalveolar lavage was performed, though only pigs immunized with 10^11^vp of Epigraph showed a significant reduction compared to unimmunized pigs (Figure 5F). In contrast to the Cluster IV(A) and Cluster I challenges, all pigs had similar levels of viral shedding from nasal secretions, and none of the groups had a statistically significant reduction in infectious virus collected from the nose (Figure 5G and Appendix A). These data indicate that only the highest dose of the Epigraph vaccine can significantly protect against infection with a highly divergent IAV-S from the 2010.1 human-like cluster.

## 4. Discussion

Every year, over 700 million hogs are produced for worldwide pork consumption. Pork production is a major source of revenue for worldwide economies. Despite this, there are still substantial economic losses due to IAV-S infection. A driving factor in a pork producer’s attitude and willingness to pay for vaccination is the benefit of a vaccine in relation to its cost and return on investment. Consequently, there are two main advantages of vaccination: (1) reduction in clinical disease to minimize economic losses to the pork industry; and (2) reduced viral shedding to decrease transmission to susceptible pigs in an infected herd. The current literature on IAV-S vaccination effectiveness in the field and its impact on production parameters is limited. However, a recent study assessing the cost versus benefit associated with porcine reproductive and respiratory virus (PRRSV) vaccination estimates that mass vaccination of sows can result in an ~USD 160–USD 290 increase in profit per sow vaccinated. Further, mass vaccination of both sows and piglets in a herd can result in an increased profit of ~USD 184–USD 368 per animal [35]. While this type of cost-benefit analysis has not been largely defined for IAV-S, it is likely that similar trends will be seen when strict vaccine programs are implemented. Consequently, an ideal IAV-S vaccine would induce protective immunity against a broad range of genetically diverse IAV-S, provide long-lasting immunity, and be inexpensive to produce to increase vaccine adherence. In the present study, we aimed to expand on our previous studies assessing the efficacy of the Epigraph vaccine and establish the minimum dose required to maintain protective responses against antigenically divergent IAV-S in swine. We observed that all doses of Epigraph protected swine against pathological damage and viral shedding after challenge with a closely matched IAV-S. We further observed that pigs vaccinated with the highest dose and second-highest dose of the Epigraph vaccine were protected against a more genetically divergent Cluster I IAV-S. Finally, we observed that pigs vaccinated with the highest dose of the Epigraph vaccine were significantly protected against challenges with a highly divergent 2010.1 cluster IAV-S compared to unimmunized pigs. These data lay the groundwork for additional development of the Epigraph vaccine as a universal H3N2 IAV-S vaccine.

A major requirement of a swH3 universal IAV-S vaccine is the induction of cross-reactive antibody responses against genetically and antigenically distinct IAV-S. After a prime or boost immunization, we observed dose-dependent antibody responses, where the highest dose of Epigraph provided the highest antibody responses and the lowest dose induced the lowest levels of antibody responses. However, we also observed minimally appreciable differences between the high-dose and medium-dose antibody responses. This indicates that decreasing the vaccine dose 10-fold had minimal effects on antibody induction. We further assessed T cell responses against three human H3N2 strains. Previous studies have demonstrated that IFN-γ-secreting T cells can protect against challenge in the absence of neutralizing antibody responses [36], highlighting the importance of T cell responses after vaccination in swine. In accordance with previous studies [18], we observed that FluSure XP induced minimal T cell responses, while all doses of the Epigraph vaccine induced cross-reactive T cell responses against the strains analyzed. Similar to the observed antibody responses, we showed a dose-dependent relationship between vaccine dose and total T cell response. Interestingly, we found that only pigs vaccinated with the highest dose of the Epigraph vaccine showed boosted T cell responses after a second immunization. This may suggest that the medium-dose and low-dose Epigraph vaccines were subjected to vector-neutralizing antibodies mounted after a prime immunization and may have been neutralized before substantially boosting T cell responses. An important but relatively understudied aspect of vaccine studies in pigs is the potential impact of sex-based differences. Biological sex is known to play a role in differential responses to immunization or infection in humans [37,38,39]. Additional factors such as castration, wean age, stress, and diet can contribute to or exacerbate sex-related differences in swine immune responses to infection or vaccination [40]. Consequently, we assessed sex-based differences in antibody and T cell responses by analyzing male- and female-specific responses at prime and boost immunizations. Similar to previous studies assessing PCV2 vaccination [41], we observed very minimal sex-based differences in HI antibody responses at different timepoints and against different viruses. Notably, we observed that female pigs in the low-dose Epigraph group exhibited significantly lower T cell responses after the boost immunization compared to the male pigs. Previous research has characterized that estrogen levels can differentially modulate T cell responses through cytokine regulation [42,43,44], and these lower responses in the female pigs may have been a result of the advancing sexual maturity of the pigs throughout the study. While hormonal changes could be a possible explanation, it should be noted that there were only minor differences, and the differences were only observed after boosting in the lowest dose group. It is reasonable that these differences may have little to no effect on overall vaccine efficacy. The sample size of males versus females in this study was relatively small compared to other studies and would be improved by additional studies with larger sample sizes of each sex.

An important aspect of the vaccine design is the delivery of the Epigraph vaccine with a human-adenovirus type 5 viral vector. The human adenovirus type 5 vector has previously been used in the swine animal model against foot and mouth disease [45], PRRSV [46,47,48], pseudorabies virus [49,50], and others. Importantly, the human adenovirus type 5 viral vector has been shown to be resistant to neutralizing maternally derived antibodies against IAV-S [51], resistant to vaccine-associated enhanced respiratory disease [52], and has a minimal risk of swine having preexisting immunity to the vector. Further, previous studies have characterized that adenoviral vectored vaccines can be mass-produced using large bioreactors and cost as low as USD 1.20 per dose of 5 × 10^10^vp [53]. Therefore, reducing the required dose 10-fold from 10^11^vp can have considerable impacts on the cost of vaccination and improve vaccine adherence.

We also performed three successive challenge studies using antigenically distinct but highly relevant H3 IAV-S strains. Since the introduction of the H3N2 subtype into swine populations in 1998, there has been extensive diversification of established clades and additional clades established through reverse zoonotic transmission events from humans to swine. Epidemiological reports of U.S. swine populations indicate that H3N2 belonging to Cluster IV(A) and the 2010.1 cluster made up ~83.9% of circulating IAV-S since 2010 [13]. Further, though Cluster I IAV-S was only minorly detected after the emergence of Cluster IV(A-F) IAV-S, recent use and reassortment of a LAIV vaccine encoding a Cluster I HA, A/swine/Texas/4199-1/1998, has resulted in the detection of reassortant H3 viruses with Cluster I HA genes in 2018–2022 [13,17]. Consequently, we assessed protection against experimental challenges with these three highly relevant IAV-S. We compared protection against pigs that were either unvaccinated or immunized with a commonly used commercial vaccine, FluSure XP. The challenge with a Cluster IV(A) IAV-S demonstrated that vaccination with the high, medium, and low dose Epigraph vaccine resulted in complete protection against pathological damage, viral replication in the lungs, and viral shedding compared to unimmunized controls. In contrast, when we challenged with a Cluster I IAV-S (with a matched HA to the LAIV vaccine), only the high-dose and medium-dose Epigraph vaccines resulted in a significant reduction in lung pathology, viral antigen load in the lungs, and reduced viral shedding, while the low-dose Epigraph vaccine had no significant reduction compared to unimmunized controls. Finally, a challenge with a highly divergent IAV-S from the 2010.1 cluster revealed that only pigs immunized with the high-dose Epigraph vaccine had reduced infectious virus in the lungs. These data support the efficacy of the Epigraph vaccine at different doses to induce broadly cross-protective immunity against a broad range of genetically divergent H3 IAV-S.

Here, we assessed the protective efficacy of the Epigraph vaccine in a dose de-escalation study and demonstrated that the Epigraph vaccine can induce highly cross-reactive antibody and T cell responses at low doses. We further assessed protection against challenge and compared protection against a commonly used commercial vaccine to assess the minimum dose required to maintain protection against three highly divergent IAV-S. We observed that decreasing the dose of the Epigraph resulted in lower protection against more genetically divergent IAV-S, but at the lowest dose, it could protect against similarly matched IAV-S. These data support the continued application of the Epigraph to induce broadly cross-reactive immunity in H3 IAV-S circulating in pig populations.

## 5. Conclusions

IAV-S is a significant pathogen that effects worldwide swine populations and poses a significant threat to global huamn health. Our previous studies have detailed the utility of the Epigraph vaccine to induce broadly cross-reactive and durable immunity in swine. Here, we characterize the minimum dose required to maintain protective immunity against three genetically divergent H3N2 IAV-S challenges. Our results demonstrate that higher doses of the Epigraph vaccine are required to protect against significantly genetically divergent IAV-S and that minimal sex-based differences were observed after immunization in swine.

## Figures and Tables

**Figure 1 vaccines-12-00943-f001:**
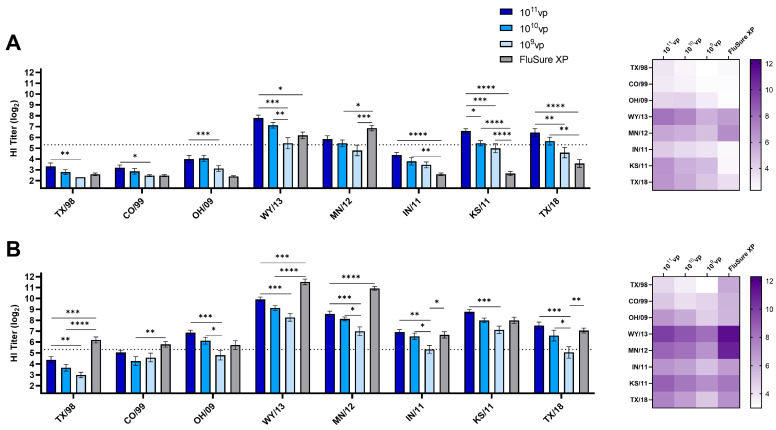
Antibody responses in dose-sparing. Male and female swine (*n* = 15/group) were immunized with FluSure XP, 10^11^vp, 10^10^vp, or 10^9^vp of Epigraph. Serum was collected to assess hemagglutination inhibition (HI) antibody responses after (**A**) prime immunization and (**B**) boost immunization. Serum from all swine was analyzed for HI antibody responses against representative IAV-S strains isolated from Cluster I-TX/98, Cluster II-CO/99, Cluster IV-OH/09, Cluster IV(A)-WY/13, Cluster IV(B)-MN/12, Cluster IV(C)-IN/11, Cluster IV(E)-KS/11, and the 2010.1 human-like cluster-TX/18. A heatmap of responses is shown on the right. The dotted line indicates an HI titer of 1:40 (5.32 log_2_). Data are presented as the mean ± SEM, and statistical analysis was determined with a one-way ANOVA with Tukey’s multiple comparisons follow-up test: * *p* < 0.05, ** *p* < 0.01, *** *p* < 0.001, **** *p* < 0.0001.

**Figure 2 vaccines-12-00943-f002:**
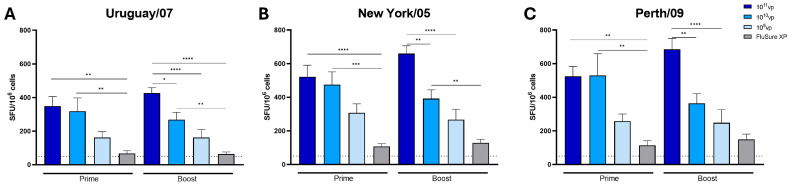
T cell responses in dose-sparing. Male and female swine (*n* = 15/group) were immunized with FluSure XP, 10^11^vp, 10^10^vp, or 10^9^vp of the Epigraph vaccine. PBMCs were collected to assess cross-reactive T cell responses by interferon-γ ELISpot after a prime and boost immunization against (**A**) A/Uruguay/716/2007, (**B**) A/New York/385/2005, and (**C**) A/Perth/16/2009. The dotted line indicates a limit of detection of 50 spot-forming units (SFU) per million cells analyzed. Data are presented as the mean ± SEM. Statistical analysis was determined with a one-way ANOVA with Tukey’s multiple comparisons follow-up test: * *p* < 0.05, ** *p* < 0.01, *** *p* < 0.001, **** *p* < 0.0001.

**Figure 3 vaccines-12-00943-f003:**
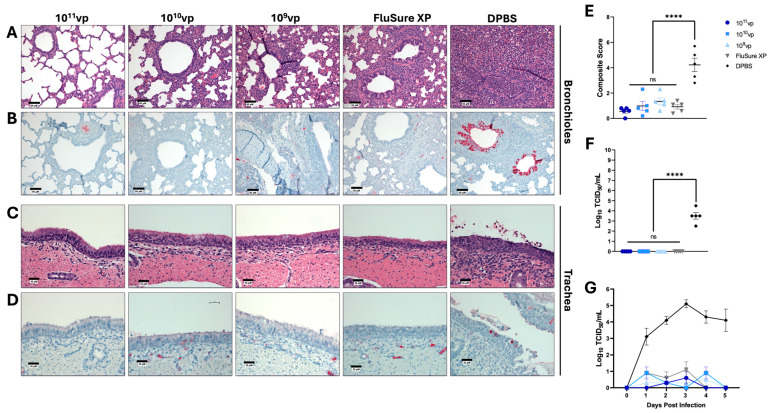
Protection against Cluster IV(A) Swine Influenza A Virus. Pigs were immunized twice with FluSure XP, sham vaccine, or 10^11^vp, 10^10^vp, or 10^9^vp of Epigraph vaccine, then challenged with 10^6^ TCID_50_ of Cluster IV(A) isolate, A/swine/Ohio/11SW87/2011. Lungs (**A**,**B**) and tracheas (**C**,**D**) were collected for histopathological analysis (**A**,**C**) and immunohistochemistry (**B**,**D**) five days post-infection. (**E**) Composite microscopic scores based on an established scoring system. (**F**) Infectious virus in the lungs was quantified using a TCID_50_ assay. (**G**) Daily nasal swabs were collected, and infectious viral shedding was quantified using a TCID_50_ assay. Data in E, F, and G are presented as the mean ± SEM, and statistical analysis was analyzed by a one-way ANOVA with Tukey’s multiple comparisons follow-up test: ns: no significance, **** *p* < 0.0001.

**Figure 4 vaccines-12-00943-f004:**
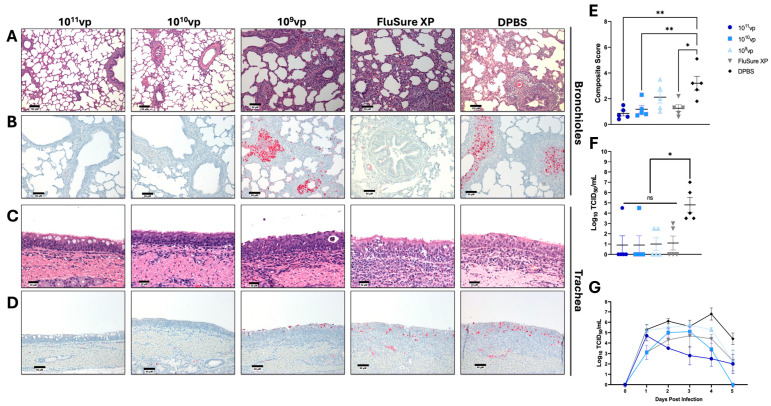
Protection against Cluster I Swine Influenza A Virus. Pigs were immunized twice with FluSure XP, sham vaccine, or 10^11^vp, 10^10^vp, or 10^9^vp of Epigraph vaccine, then challenged with 10^6^ TCID_50_ of Cluster I isolate, A/swine/Texas/4199-2/1998. Lungs (**A**,**B**) and tracheas (**C**,**D**) were collected for histopathological analysis (**A**,**C**) and immunohistochemistry (**B**,**D**) five days post-infection. (**E**) Composite microscopic scores based on an established scoring system. (**F**) Infectious virus in the lungs was quantified using a TCID_50_ assay. (**G**) Daily nasal swabs were collected, and infectious viral shedding was quantified using a TCID_50_ assay. Data in E, F, and G are presented as the mean ± SEM, and statistical analysis was analyzed by a one-way ANOVA with Tukey’s multiple comparisons follow-up test: ns: no significance, * *p* < 0.05, ** *p* < 0.01.

**Figure 5 vaccines-12-00943-f005:**
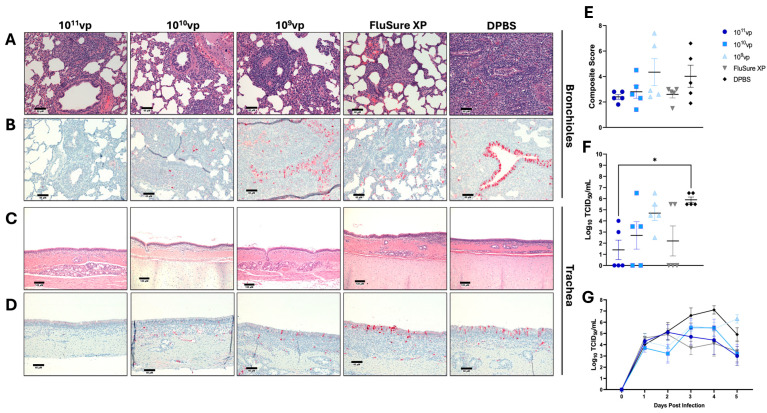
Protection against Cluster 2010.1 Human-like Swine Influenza A Virus. Pigs were immunized twice with FluSure XP, sham vaccine, or 10^11^vp, 10^10^vp, or 10^9^vp of Epigraph vaccine, then challenged with 10^4.5^ TCID_50_ of Cluster 2010.1 human-like isolate, A/swine/Texas/A01785781/2018. Lungs (**A**,**B**) and tracheas (**C**,**D**) were collected for histopathological analysis (**A**,**C**) and immunohistochemistry (**B**,**D**) five days post-infection. (**E**) Composite microscopic scores are based on an established scoring system. (**F**) Infectious virus in the lungs was quantified using a TCID_50_ assay. (**G**) Daily nasal swabs were collected, and infectious viral shedding was quantified using a TCID_50_ assay. Data in E, F, and G are presented as the mean ± SEM, and statistical analysis was analyzed by a one-way ANOVA with Tukey’s multiple comparisons follow-up test: * *p* < 0.05.

## Data Availability

All data relevant to the study are available in the main figures or the Appendix A.

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
