# Peer review of "Immunogenicity and Protective Efficacy of Dose-Sparing Epigraph Vaccine against H3 Swine Influenza A Virus"

_vaccines, 2024, doi:10.3390/vaccines12080943_

Round 1

Reviewer 1 Report

Comments and Suggestions for Authors

The manuscript (ID:3150794) by Erika Petro-Turnquist et al. was to explore the immunogenicity and protective efficacy of dose-sparing Epigraph vaccine against H3 swine influenza A virus. The authors revealed that a dose-dependent effect where the highest dose of Epigraph protected against all three challenges, the middle dose Epigraph protected against more genetically similar IAV-S, and the lowest dose Epigraph only protected against genetically similar IAV-S. The results of these studies can be used to continue developing a broadly protective and low-dose vaccine against H3 IAV-S.

The followings should be improved:

1. In experiment 3, Haemophilus parasuis was detected in 9 out of 25 pigs at necropsy. Please explain whether Haemophilus parasuis infection disturbs the results such as T cell response in experiment 3? why?

2. In the section of “Lung Pathological Analysis”, the experimental procedure of immunohistochemistry should be improved .

3. The degree of pathological change in the lung and trachea tissue sections was scored  according to a previously described scoring system. Please provide the scoring system.

4. The abbreviation should be checked in the whole manuscript, such as “Peripheral blood mononuclear cells (PBMCs) “ appears several times in the manuscript, please check in the whole manuscript.

5. In Figure 1 legends, “(A) or (B)”were appeared two times, which can not accurately express the author's meaning.

6. The images about HE and IHC in figure 3-5 need to provide magnification or size standards.

7. The levels of IFN-γ secreting T cell responses against vaccine are different in males and females pigs, why? Please discuss it.

Comments on the Quality of English Language

The manuscript (ID:3150794) by Erika Petro-Turnquist et al. was to explore the immunogenicity and protective efficacy of dose-sparing Epigraph vaccine against H3 swine influenza A virus. The authors revealed that a dose-dependent effect where the highest dose of Epigraph protected against all three challenges, the middle dose Epigraph protected against more genetically similar IAV-S, and the lowest dose Epigraph only protected against genetically similar IAV-S. The results of these studies can be used to continue developing a broadly protective and low-dose vaccine against H3 IAV-S.

The followings should be improved:

1. In experiment 3, Haemophilus parasuis was detected in 9 out of 25 pigs at necropsy. Please explain whether Haemophilus parasuis infection disturbs the results such as T cell response in experiment 3? why?

2. In the section of “Lung Pathological Analysis”, the experimental procedure of immunohistochemistry should be improved .

3. The degree of pathological change in the lung and trachea tissue sections was scored  according to a previously described scoring system. Please provide the scoring system.

4. The abbreviation should be checked in the whole manuscript, such as “Peripheral blood mononuclear cells (PBMCs) “ appears several times in the manuscript, please check in the whole manuscript.

5. In Figure 1 legends, “(A) or (B)”were appeared two times, which can not accurately express the author's meaning.

6. The images about HE and IHC in figure 3-5 need to provide magnification or size standards.

7. The levels of IFN-γ secreting T cell responses against vaccine are different in males and females pigs, why? Please discuss it.

Reviewer 2 Report

Comments and Suggestions for Authors

Dear authors,

Thank you for the manuscript, I read it with interest. The results presented in the paper are of high interest to swine producers, veterinary and influenza professionals.  There are a few remarks that I would like to be considered:

1)      P.2 line 83, please check the sentence, it seems the word “antibodies” is missing after cross-protective

2)      P.3 line 99, please check “swine were house” , seems to be mistake

3)      In 2.2. you list the viruses. Please include the sentence about subtype (all H3N2 I guess) and it is important to include information about genetic lineage of each strain here as well (including GISAID or GenBank accession numbers)

4)      In 2.3 please include the information on the volume of virus-containing fluid that was used to challenge pigs, also in 106 TCID50indicate the volume per which the TCID50 was determined (per ml or 0,2 ml); indicate subtype and genetic group for 11SW87 challenge virus and please provide explanation why the lower inoculation dose was used for swine/Texas/A01785781/2018 virus

5)      In 2.4. please explain why do you choose chicken red blood cells to perform HAI  although all viruses were MDCK-grown. This could have affected the HAI results

6)      In 2.5. please subtype and genetic group for the viruses listed; given these are human influenza viruses add information about their relatedness with circulating swine influenza strains

7)      3.1. please provide at least brief description of FluSure XP vaccine for the reader. Otherwise it is difficult to understand why this vaccine is used as a control

8)      It’s recommended to enlarge Figure 1, currently it’s too small to see all the details

9)      Discussion, line 436 – it is not a universal vaccine, rather the swH3 universal vaccine, as it was already indicated in one of your earlier papers

Reviewer 3 Report

Comments and Suggestions for Authors

In this study, the authors assessed the immunogenicity and protective efficacy of the Epigraph vaccine at increasingly lower doses to determine the minimum dose required to maintain protective immunity against three genetically divergent H3 IAV-S. However, there are some points need to be concerned.

1.     How did the protection threshold determined in "3.1. Antibody Responses in Dose Sparing"?

2.     The scale in the pathology section images of this paper is unclear; the images should be revised accordingly.

3.     This study found that the effectiveness of immune protection is related to the immunization dose. Commercial vaccines were used as a control, but what is the antigen content of the commercial vaccines?

4.     The commercial vaccine induces significantly lower levels of IFN-γ compared to the Epigraph vaccine but still provides good protection. What is the significance of measuring IFN-γ in this experiment? Additionally, in "3.2 T Cell Responses in Dose Sparing," it is necessary to reference the specific antibody levels induced by the respective strains when immunizing with different strains.

5.     There are some errors in the reference formatting that need to be corrected.

Comments on the Quality of English Language

  The language in this paper is not standardized and is overly colloquial; the entire text should be revised for formal description.
